# Comparative Metabolomic Analysis Reveals Distinct Flavonoid Biosynthesis Regulation for Leaf Color Development of *Cymbidium sinense* ‘Red Sun’

**DOI:** 10.3390/ijms21051869

**Published:** 2020-03-09

**Authors:** Jie Gao, Rui Ren, Yonglu Wei, Jianpeng Jin, Sagheer Ahmad, Chuqiao Lu, Jieqiu Wu, Chuanyuan Zheng, Fengxi Yang, Genfa Zhu

**Affiliations:** Guangdong Key Laboratory of Ornamental Plant Germplasm Innovation and Utilization, Environmental Horticulture Research Institute, Guangdong Academy of Agricultural Sciences, Guangzhou 510640, China; gaojie@gdaas.cn (J.G.); Renruinjau@163.com (R.R.); hjyylab@126.com (Y.W.); 13424050551@163.com (J.J.); sagheerhortii@gmail.com (S.A.); chuqiaolu18@163.com (C.L.); jiebaoqiqiqiu@163.com (J.W.); 15014323983@163.com (C.Z.)

**Keywords:** metabolomic analysis, differential metabolites, enzyme activity, gene expression, leaf color, *Cymbidium sinense*

## Abstract

The colorful leaf is an important ornamental character of *Cymbidium sinense* (*C. sinense*), especially the red leaf, which has always been attracted by breeders and consumers. However, little is documented on the formation mechanism of the red leaf of *C. sinense*. In this study, the changing patterns of flavonoid-related metabolites, corresponding enzyme activities and genes expression in the leaves of *C. sinense* ‘Red Sun’ from red to yellow and finally to green was investigated. A total of 196 flavonoid-related metabolites including 11 anthocyanins metabolites were identified using UPLC-MS/MS-based approach. In the process of leaf color change, 42 metabolites were identified as having significantly different contents and the content of 28 differential metabolites turned to zero. In anthocyanin biosynthetic pathway, content of all 15 identified metabolites showed downregulation trend in the process of leaf color change. Among the 15 metabolites, the contents of Naringenin chalcone, Pelargonidin O-acetylhexoside and Anthocyanin 3-O-beta-d-glucoside decreased to zero in the green leaf stage. The changing pattern of enzyme activity of 10 enzymes involved in the anthocyanin biosynthetic pathway showed different trends from red leaves that have turned yellow and finally green, while the expression of genes encoding these enzymes was all down-regulated in the process of leaf color change. The results of this study revealed the types of flavonoid-related metabolites and the comprehensive analysis of metabolites content, enzyme activities and genes expression providing a new reference for breeders to improve the leaf color of *C. sinense* ‘Red Sun’.

## 1. Introduction

As the largest family of monocotyledons, Orchidaceae has a long history of cultivation and contains abundant varieties. Among these varieties, *C. sinense* is a unique variety produced in China, which has been loved by consumers all around the world. The most important phenotypic feature of *C. sinense* is that it has a very rich variation in leaf color, which improves its ornamental and economic values. *C. sinense* contains a number of varieties based on the location and color of leaf variegation, including chimera art, claw art, crown art, crane art, middle penetration art, spot art and treasure art and so on [1]. At present, the leaf variegation varieties of *C. sinense* are basically yellow-green leaves, but there are few reports about red leaves. *C. sinense* ‘Qihei’ is the most common variety of green leaves. In the process of culture, the color of ‘Qihei’ leaf buds changes to red, and then the leaf color changes from red to yellow, finally to green, leaving only a little yellow at the tip of the leaf during the growth and development of leaf buds. The leaf variegation variety is named *C. sinense* ‘Red Sun’. So far, the metabolic changes associated with the formation of red leaves in *C. sinense* ‘Red Sun’ are not known.

Compared with flower color, the influencing factors of leaf color are more complex. Generally, the greening of leaves is mainly due to the absolute proportion of chlorophyll, and the formation of yellow leaves is mostly due to the degradation of chlorophyll, leaving the color of carotenoids to dominate in leaves [1,2]. Anthocyanins are responsible for the red color of leaves, which is helpful for plants against various biotic and abiotic stresses [3,4]. In *Cymbidium* orchids, the activation of anthocyanin synthesis can be restored by introducing MYB and bHLH anthocyanin regulators simultaneously [5]. The synthesis pathway of anthocyanins begins with the precursor phenylalanine, resulting in the formation of dihydrokaempferol (DHK) by the catalysis of six enzymes. After the catalysis of four enzymes, DHK produces three kinds of steady-state anthocyanins, including pelargonidin, cyanidin and delphinidin, respectively [6,7,8]. At present, more than 600 anthocyanins found in nature are derived from these three substances and the accumulation of anthocyanins in plants has two main functions: one is to produce rich and colorful visual signals to promote pollination or seed transmission, and the other is to resist a series of biotic or abiotic stresses [9,10].

Plant metabolomics carries out a qualitative and quantitative analysis of small molecular metabolites in plants, so as to help researchers understand the synthesis and accumulation patterns of metabolites [11]. At present, the study of plant metabolites mainly involves the identification of metabolites, variety differentiation and auxiliary breeding and so on [12,13,14,15]. Flavonoids-targeted metabolomics refers to the analysis of small molecular metabolites of flavonoids (anthocyanins), which is often used to analyze the formation mechanism of plant color. Analysis of Flavonoids-targeted metabolomics in the white and purple flowers of *Phalaenopsis* identified 142 different flavonoid-related metabolites, of which the most important anthocyanin was the derivative of cyaniding [16]. A study on green and purple asparagus suggested that the difference in color was mainly caused by the contents of peonidin and cyanidin and their glycoside derivatives [17]. Metabolomic analysis the color difference of fig peel showed that the content of anthocyanin derivatives in purple peel was significantly higher than that in green fruit [18]. These results indicate that metabolomics is an important and effective method to analyze the formation mechanism of plant color.

In order to understand the key metabolites involved in the process of color change from red to green in the leaves of *C. sinense* ‘Red Sun’, flavonoids-targeted metabolomic analysis was used to analyze the change pattern of flavonoid-metabolites in the process of leaf color change. Physiology, enzyme kinetics and molecular biology experiments were carried out to explore the mechanism of leaf color difference. To the best of our knowledge, this study is the first effort to analyze the mechanism of the formation of *C. sinense* leaf variegation from the perspective of metabolites, and the results provide a new train of thought and basis for the study of *C. sinense* leaf variegation.

## 2. Results

### 2.1. Pigments Content Analysis

The leaves of *C. sinense* ‘Red Sun’ show red at the leaf bud stage. With the development of the leaves, the leaves gradually change from red to yellow, and finally the mature leaves develop into green with a little yellow color at the tip, as seen in Figure 1. The color of leaves was affected by a variety of pigments, including the contents of chlorophyll, carotenoids, total flavonoids and total anthocyanins in *C. sinense* ‘Red Sun’. The results showed that the maximum contents of the four pigments were found at the red leaf stage, while the minimum contents were shown by leaves at yellow stage. The contents of all four pigments in green leaves were higher than those in yellow leaves, but lower than those in red leaves (Figure 2).

### 2.2. Qualitative and Quantitative Analyses of Metabolites and Quality Control (QC) Analysis of Sample

Anthocyanin is the cause of red leaf color. From the pigment contents, it can be seen that the contents of total flavonoids and total anthocyanins change significantly in the process of leaf color change. To compare the differences of flavonoid-metabolites, the UPLC (Shim-pack UFLC SHIMADZU CBM30A)−MS/MS (Applied Biosystems 4500 QTRAP) technique was used to detect the flavonoid-related metabolites in RL, YL and GL. The total ion flow map of the mixed sample quality control (QC) sample (Total ions current, TIC, is the map obtained by adding the intensities of all ions in each time point mass spectrum) is shown in Appendix A. The multi-peak map of MRM metabolite detection of multi-substance extraction is illustrated in Appendix A. Based on the local metabolic database, the metabolites of the samples were qualitatively and quantitatively analyzed by mass spectrometry. The multi-peak map of MRM metabolite detection in the multi-reaction monitoring mode shows the substances that can be detected in the sample, and each mass peak of different color represents a detected metabolite. A total of 196 flavonoid-related metabolites were detected (Appendix A), including 11 anthocyanins, 3 Chalcone, 9 Dihydroflavonoid, 5 Dihydroflavonol, 12 flavanols, 11 Flavone, 59 Flavonoid, 20 Flavonoidcarbonoside, 3 Flavonoids, 59 Flavonols and 4 Isoflavones. In the process of instrumental analysis, one QC sample is inserted into every 10 test and analysis samples to monitor the repeatability of the analysis process. Through the overlap display analysis of the TIC map of different quality control QC samples (Appendix A), the results showed that the curve overlap of metabolite detection of total ion current was high, that is, the retention time and peak intensity were the same, indicating that the signal stability was good when the same sample was detected by mass spectrometry at different time points. All the detected metabolite content data were normalized by range method, and the accumulation patterns of metabolites among different samples were analyzed by cluster analysis (Hierarchical cluster analysis, HCA) by R software (www.r-project.org/) (Figure 3).

### 2.3. Formatting of Mathematical Components Principal Component Analysis (PCA) and Orthogonal Projections to Latent Structures-Discrimination Analysis (OPLS-DA)

Principal Component Analysis (PCA) is a multidimensional data statistical analysis method of unsupervised pattern recognition. Through principal component analysis of samples (including QC samples), we can preliminarily understand the overall metabolic differences among samples and the degree of variability between samples within groups. From the analysis results, it can be observed that there are significant differences among RL, YL and GL groups, but there is no significant difference within groups (Appendix A).

Although PCA can effectively extract the main information, it is not sensitive to variables with small correlation, and Partial Least Squares-Discriminant Analysis (PLS-DA) can solve this problem. Compared with PCA, PLS-DA can maximize the distinction between groups and facilitate the search for differential metabolites. Through the analysis of PLS-DA, the orthogonal variables, which are not related to the classification variables of metabolites are first eliminated, and then the differences of correlation between groups and within groups are analyzed. According to the OPLS-DA model, we analyzed the metabolic group data, draw the score chart of each group, and further showed the differences between each group. The prediction parameters of the evaluation model are R2X, R2Y and Q2, in which R2X and R2Y represent the interpretation rate of the model to X and Y matrix respectively, and Q2 indicates the prediction ability of the model. The closer these three indexes are to 1, the more stable and reliable the model is. Q2 > 0.5 can be regarded as an effective model, and Q2 > 0.9 is an excellent model. From the results, there are significant differences among the three groups of data, but there is no significant difference between groups (Appendix A). The alignment verification of OPLS-DA was carried out (*n* = 200, that is, 200 permutation experiments were carried out). In the model verification, the horizontal lines correspond to R2 and Q2 of the original model, and the red dots and blue dots represent R2’ and Q2’ of the model after Y replacement, respectively. The results showed that R2’ and Q2’ of each group were smaller than R2 and Q2 of the original model, which indicated that the model was meaningful and the differential metabolites could be screened according to VIP value analysis (Appendix A).

### 2.4. Screening Differential Metabolites in the Process of Leaf Color Change

The differential metabolites were screened by combining the VIP values of fold change and OPLS-DA model. Following was the screening criteria: (1) If the difference of metabolites content between the control group and the experimental group is more than 2 times or less than 0.5, the difference is considered to be significant; (2) On the basis of the above, the metabolites with VIP ≥ 1 are selected. Volcano Plot was used to show the difference of expression level between the two groups of samples, and the difference was statistically significant (Figure 4). Compared with red leaves, 85 metabolites in green leaves were significantly different (10 up-regulated and 75 down-regulated), and 61 metabolites in green leaves were significantly different from yellow leaves (16 up-regulated and 45 down-regulated) (Appendix A). Compared with red leaves, 86 metabolites showed significant change in yellow leaves (3 up-regulated and 83 down-regulated) (Appendix A). Through the analysis of Vennn diagram, it was found that there were 42 metabolites with different contents in the three periods of leaf color change. Except for the content of Liquiritinapioside, the other 41 substances decreased significantly with the change of leaf color (Appendix A). After a comprehensive analysis of the contents of all the differential metabolites, we found that the contents of 28 metabolites turned to zero in the process of leaf color change, as shown in Figure 5. Among these metabolites with drastic changes, the contents of Naringenin chalcone, an important intermediate in anthocyanin synthesis, and two important anthocyanin metabolites, Pelargonidin O-acetylhexoside and Anthocyanin 3-O-beta-D-glucoside, all decreased to zero in the green leaf stage.

### 2.5. Intermediates Content, Enzymes Activities and Genes Expression Associated with the Anthocyanin Biosynthetic Pathway

In order to further analyze the regulation mechanism of anthocyanin in the process of *C. sinense* ‘Red Sun’ leaf color change, the detected intermediates of anthocyanin synthesis pathway and all the enzyme activities involved in anthocyanin synthesis were analyzed [19] (Figure 6). It can be observed that all 15 detected intermediates show a downward trend in the process of leaf color change. In total, 11 kinds of anthocyanins metabolites were detected, and the contents of eight of them (Cyanidin 3-rutinoside, Cyanidin 3-O-galactoside, Peonidin 3-O-glucoside chloride, Cyanidin 3-O-malonylhexoside, Cyanidin O-acetylhexoside, Anthocyanin 3-O-beta-d-glucoside, Pelargonidin 3-O-malonylhexoside and Pelargonidin O-acetylhexoside) changed significantly during the change of leaf color, especially the content of Pelargonidin O-acetylhexoside (molecular weight: 474.096 Da), a derivative of Pelargonidin, and Anthocyanin 3-O-beta-d-glucoside (molecular weight: 449.089 Da), a derivative of Cyanidin finally decreased to 0 in green leaves. After measuring the activity of all the enzymes involved in anthocyanin synthesis, it was found that the activity of henylalanine ammonia-lyase (PAL), trans-cinnamate 4-monooxygenase (C4H), 4-coumarate-CoA ligase (4CL) and flavonoid 3’,5’-hydroxylase (F3′5′H) was up-regulated, the activity of chalcone synthase (CHS), chalcone isomerase (CHI), naringenin 3-dioxygenase (F3H) and flavonoid 3’-monooxygenase (F3′H) was down-regulated. The activity of DFR was down-regulated at first and then up-regulated, while the activity of ANS was up-regulated and then decreased. QRT-PCR analysis of the genes encoding these enzymes showed that all genes were significantly down-regulated (Figure 7). Comprehensive analysis of enzyme activity and the corresponding gene expression pattern showed that the changing trend of enzyme activity of CHS, CHI, F3H, F3’H was consistent with that of gene expression pattern, while that of the other six enzymes was different from that of gene expression pattern.

## 3. Discussion

The regulation of plant leaf color is a complex process. Most of the previous studies analyzed the mechanism of leaf color formation by methods of physiology, cytology and molecular biology [1], but the mechanism of leaf color regulation from the perspective of small molecular metabolites needs to be studied further. In this study, based on UPLC-MS/MS, the changes of metabolites in the process of leaf color change of *C. sinense* ‘Red Sun’ were qualitatively and quantitatively analyzed, and the unique pattern of flavonoid-related metabolites in the process of leaf color change was constructed for the first time. At the same time, the regulation mechanism of leaf color was further analyzed by enzyme kinetics and gene expression analysis.

As far as we know, this is the first time to analyze the types of flavonoid-related metabolites in the leaves of *C. sinense* ‘Red Sun’ by UPLC-MS/MS (flavonoids-targeted) method. A total of 196 flavonoid-related metabolites were detected. These substances belong to anthocyanin, chalcone, dihydroflavonoid, dihydroflavonol, flavanols, flavone, flavonoid, flavonoid carbonoside, flavonoids, flavonols and isoflavones. Based on wide target metabolomics analysis, only 6 and 15 differential flavonoid-related metabolites were detected in tea leaves and *Ginkgo biloba* leaves, respectively [20,21]. Based on phenolic-targeted secondary metabolites analysis in purple fig peel, only 15 differential flavonoid-related metabolites (including four anthocyanins metabolites) were detected [18]. In this study, 119 kinds of differential flavonoid-related metabolites were found (including 10 kinds of anthocyanins). The above results show that Flavonoids-targeted metabolites method can identify more kinds of flavonoid-related metabolites, and has more advantages in mining the types and contents of flavonoid-related metabolites, especially anthocyanins metabolites.

*Phalaenopsis* and the materials of this study belong to Orchidaceae. In the metabolomic analysis between petals of white and purple *Phalaenopsis*, 142 differential flavonoid-related metabolites, including 17 anthocyanins metabolites, were detected by flavonoids-targeted metabolomic analysis [16]. In accordance with the results of this research, among the 119-differential flavonoid-related metabolites detected, there were 8 differential anthocyanins. The differential metabolites of eight anthocyanins identified in the leaves of *C. sinense* ‘Red Sun’ are Cyanidin 3-rutinoside (Keracyanin chloride), Cyanidin 3-O-galactoside, Peonidin 3-O-glucoside chloride, Cyanidin 3-O-malonylhexoside, Pelargonidin 3-O-malonylhexoside, Cyanidin O-acetylhexoside, Pelargonidin O-acetylhexoside and Anthocyanin 3-O-beta-d-glucoside. Compared with the 18 anthocyanin differential metabolites detected in the petals of *Phalaenopsis* [16], only Cyanidin 3-O-malonylhexoside and Cyanidin O-acetylhexoside are the same. Among the three main categories of pigmented glycosides, pelargonidin mainly shows orange/red, cyaniding mainly shows pink/magenta and delphinidin mainly shows purple/blue [22,23]. The metabolites with the highest proportion were cyanidin derivatives found in the petals of *Phalaenopsis* and leaves of *C. sinense* ‘Red Sun’. Interestingly, during the process of *C. sinense* ‘Red Sun’ leaf color change, the two anthocyanins from existence to absence are Pelargonidin 3-O-malonylhexoside and Anthocyanin 3-O-beta-D-glucoside; the content of derivatives of six kinds of cyanidin and a kind of delphinidin was high in purple petals but zero in white petals of *Phalaenopsis* [16]. The above results are consistent with the phenotype of the corresponding materials.

As the upstream reaction of anthocyanin and other flavonoids, Phenylalanine was first converted to P-coumaroyl-CoA under the catalysis of PAL, C4H and 4CL [24]. The activities of PAL, C4H and 4CL were all up-regulated during the change of leaf color, while the coding genes expression of these three enzymes decreased significantly, which indicated that the activities of these three enzymes were also subject to post-transcriptional modification or post-translational modification [25]. Then P-coumaroyl-CoA was transformed into Naringenin under the action of CHS and CHI [24]. The activities of CHS and CHI were down-regulated in the process of leaf color change, which was consistent with the change pattern of gene expression. CHS is the first key enzyme in anthocyanin synthesis, its activity determines the formation of anthocyanin metabolic pathway, and the loss of its activity will lead to the loss of anthocyanin and other flavonoids [26,27]. From the results of qRT-PCR, the expression of CHS-3 in the red leaf stage was 813 times higher than that in the green leaf stage, indicating that CHS may play an important role in the process of leaf color change. Overall, similar correlations between gene expression and anthocyanin levels were also observed during the differential pigment deposition in crabapple cultivars with dark red, pink and white petal colors [28]. Next, Naringenin forms DHK under the catalysis of F3H. DHK then forms Dihydroquercetin (DHQ) and Dihydromyricetin (DHM) under the catalysis of F3′H and F3′5′H, respectively [29]. The enzyme activity of F3H and F3′H was down-regulated, while that of F3′5′H was up-regulated, and the coding gene expression of these three enzymes was significantly down-regulated. Both natural mutants and transgenic studies have proved that the competitions of three enzymes lead to different branching pathways at this critical point [30], and our results support this argument. DHK, DHQ and DHM formed unstable anthocyanins under the catalysis of DFR and ANS. The enzyme activity of DFR decreased at first and then increased, while the activity of ANS increased at first and then decreased, but the coding genes expression level of the two enzymes was significantly down-regulated. DFR from different plants has specific substrates biases for DHK, DHQ and DHM [6,31], and the downstream DFRs and ANSs is necessary for large sum of anthocyanin accumulation in *Phalaenopsis* [32,33]. These unstable anthocyanins eventually went through the action of UFGT to form stable anthocyanins [34]. During the period of color change, UFGT activity firstly decreased, and then increased, while the *UFGT* gene expression level was significantly down-regulated (Appendix A). Previous studies have found that overexpression of UFGT causes plants to show darker colors, such as crimson or purple, while overexpression of DFR or ANS only deepens the color to pink or lavender [34,35,36]. In this study, the results of qRT-PCR showed that the expression of ANS in red leaf stage was 833 times higher than that in green leaf stage, while the expression of UFGT in red leaf stage was 14 times higher than that in green leaf stage. The question of whether ANS or UFGT had a greater effect on the leaf color of *Cymbidium* remains to be further verified.

## 4. Materials and Methods

### 4.1. Plant Materials

The three-year-old *C. sinense* ‘Red Sun’ planted in the glass greenhouse located in Environmental Horticulture Research Institute of Guangdong Academy of Agricultural Sciences was used as the research material. According to the change pattern of leaf color, red leaf samples of three independent plants (RL1, RL2 and RL3) were taken on 15 January 2019, yellow leaf samples of three independent plants (YL1, YL2 and YL3) were taken on 25 February 2019 and green leaf samples of three independent plants (GL1, GL2 and GL3) were taken on 21 March 2019. The samples were quickly fixed with liquid nitrogen and stored at −80 °C.

### 4.2. Pigments Content Measurement

The contents of total chlorophyll and carotenoids were determined by spectrophotometric analyses. About 0.1 g leaves (accurately record the weight) were cut into pieces and rinsed with distilled water. Add 1 mL extracting solution (95% ethanol) and 50 mg calcium carbonate powder in mortar, the mixtures were grinded fully in low light conditions and transferred to 10 mL glass test tube. The mortar was rinsed with the extracting solution, all the washing solution was transferred into the glass tube and replenished to 10 mL with the extracting solution. Keep the glass tube in the dark until the tissue is completely whitened. Add 200 μL leach liquor in a 96-well plate, the automatic microplate reader (Sunrise, TECAN, Switzerland) was set to zero by extracting solution, then the absorbance values of 663 nm, 645 nm and 470 nm were determined and recorded as A663, A645 and A470, respectively. The calculation formula is as follows: total chlorophyll content (mg/g) = 0.02 × (20.21 × A6458.02 × A663) × N/M, total carotenoid concentration (mg/g) = 0.02 × [(1000A470–3.27Ca–104Cb)/229] × N/M (N represents dilution multiple, M represents sample fresh quantity).

The content of total anthocyanins was determined by colorimetry method. The sample was dried in the oven (37 °C) to a constant weight and crushed. After passing through a 40-mesh sieve, about 0.1 g sample was weighed (accurately recording the weight). Add 1 mL extracting solution (95% ethanol: 1.5 mol/L HCL = 85:15) in samples and the mixtures were extracted at 4 °C for 24 h. Then the mixtures were centrifuged at 8000× *g* for 10 min at room temperature, and the supernatant was taken to be tested. The automatic microplate reader (Sunrise, Switzerland) was preheated more than 30min, and reagent A (PH 1.0 buffer, 0.2mol/L KCL : 0.2mol HCL = 25:67) and reagent B (PH 4.5 buffer, 1 mol NaAC : 1 mol HCL: H_2_O = 100:60:90) were preheated more than 10 min. Add 180 μL reagent A to 20 μL supernatant and let stand for 15 min, then the absorbance values at 530 nm and 700 nm were determined and were recorded as A1 and A2, respectively. Add 180 μL reagent B to 20 μL supernatant and let stand for 15 min, then the absorbance values at 530 nm and 700 nm were determined and were recorded as A3 and A4, respectively. The calculation formula is as follows: total anthocyanin concentration (µg/g) = 33.4 × [(A1 − A2) − (A3 − A4)] × N/M (N represents dilution multiple; M represents sample dry weight, g).

The content of total flavonoids was determined by colorimetry method. The sample was dried in the oven (37 °C) to a constant weight and crushed. After passing through a 40-mesh sieve, about 0.1 g sample was weighed (accurately recording the weight). Add 1 mL extracting solution (60% ethanol), the mixtures A were extracted by ultrasonic method (ultrasonic power: 300 W, crushing time: 5 s, interval time: 8 s) at 60 °C for 30 min. Then, the mixtures were 12,000 rpm for 10 min at room temperature, and the supernatant A was replenished to 1 mL by extracting solution and taken to be test. The automatic microplate reader (Sunrise, Switzerland) was preheated more than 30 min, set to 470 nm and zero by distilled water. Reagent A (5% nitrous acid), reagent B (10% aluminum nitrate), reagent C (4% sodium hydroxide solution) and standard (10 mg/mL tannic acid standard solution) were prepared. The supernatant was treated according to the instruction manual of the kit for the determination of total flavonoids in plants (Appendix A). The mixtures B was mixed well and placed in a water bath at 37 °C for 45 min. Then the mixtures B was centrifuged at 10,000× *g* for 10 min at room temperature, and 200 uL supernatant B was used to determine absorbance values at A470 in a 96-well plate. The calculation formula is as follows: the flavonoids concentration (mg/g) = C × N × V/M (C represents the content of flavonoids observed in the standard curve, mg/mL; N represents dilution multiple; V represents the total volume of extracting solution, ml; M represents the dry weight of sample, g). 

Flavonoids-targeted metabolomics analysis was performed by liquid chromatography-mass spectrometry (LC-MS) at Metware Biotechnology Co.,Ltd (Wuhan, China) as described by [37], with small modifications. Metabolomics analysis includes two parts: metabolomics experiment and data analysis, based on the metabolite data obtained from experimental design, sample collection and processing, metabolite extraction and metabolite detection and analysis. It can carry out the identification of metabolites and the quality control analysis of sample data, and screen out some differential metabolites, so as to predict and analyze the related functions of the metabolites of the samples.

### 4.3. Analysis of Flavonoids-Targeted Metabolomics

#### 4.3.1. Standards and Reagents

The standard (analytical purity) was purchased in BioBioPha (Kunming, China) or Sigma-Aldrich (St Louis, MO, USA), and dissolved in dimethyl sulfoxide (DMSO) or methanol as solvent and stored at −20 °C. Before mass spectrometry analysis, 70% methanol was diluted to different gradient concentrations. Methanol, acetonitrile and ethanol were all analytically pure and purchased in Merck (Darmstadt, Germany).

#### 4.3.2. Sample Extraction Process

The leaves of *C. sinense* ‘Red Sun’ were vacuum freeze-dried and ground with a grinder (30 Hz for 1.5 min) to powder. The 0.1 g powder was dissolved in 70% methanol aqueous solution. The dissolved sample was swirled three times at 4 °C overnight to improve the extraction rate. After centrifugation (rotating speed 10,000× *g*, 10min), the supernatant was filtered with a microporous membrane (0.22 μ m pore size). The sample was stored in a sample injection bottle for UPLC-MS/MS analysis.

#### 4.3.3. Acquisition Conditions of LC-MS

The data acquisition instrument system mainly includes Ultra Performance Liquid Chromatography (UPLC) (Shim-pack UFLC SHIMADZU CBM30A, http://www.shimadzu.com.cn/) and Tandem mass spectrometry (MS/MS) (Applied Biosystems 4500 QTRAP, http://www.appliedbiosystems.com.cn/).

UPLC was run under following conditions: (1) Waters ACQUITY UPLC HSS T3 C18 1.8 µm, 2.1 mm × 100 mm; (2) The aqueous phase was ultra-pure water (0.04% acetic acid is added) and the organic phase was acetonitrile (adding 0.04% acetic acid); (3) Elution gradients: 95:5 v/v at 0 min, 5:95 v/v at 11.0 min, 12 min, 12.1 min and 15 min; (4) The flow rate was 0.4 mL/min, the column temperature was 40 °C, and the injection volume was 5 μL. MS/MS worked with these conditions: (1) Curtain gas (CUR) was set to 25 psi; (2) Electrospray ionization (ESI) was set to 550 °C; (3) the MS voltage was set to 5500V; (4) Dclusteringotential (DP) was optimized; (5) Collision energy (CE) was optimized; (6) Collision-activated dissociation (CAD) was set to high.

#### 4.3.4. Data Evaluation

Based on the self-built database MWDB (metware database) and the public database of metabolite information, the primary and secondary spectral data of mass spectrometry were qualitatively analyzed by software Analyst 1.6.3. For the qualitative analysis of some substances, the interference from isotope signals are removed, including duplicate signals of K^+^, Na^+^ and NH_4_^+^, as well as duplicate signals of fragment ions which were derived from other large molecules. The structure analysis of metabolites refers to the existing mass spectrometry public databases such as MassBank (http://www.massbank.jp/), KNAPSAcK (http://kanaya.naist.jp/KNApSAcK/), HMDB (http://www.hmdb.ca/) (Wishartetal.2013), MoToDB (http://www.ab.wur.nl/moto/) and METLIN (http://metlin.scripps.edu/index.php) [38].

The quantification of metabolites was carried out by using multiple reaction monitoring (MRM) mode. In the MRM model, the quadrupole first selected the precursor ions (parent ions) of the target substance. While screening the corresponding ions of other molecular weight substances to initially eliminate the interference, the precursor ions were ionized by the collision chamber to form a lot of fragment ions. Then, the fragment ions were filtered through the triple quadrupole to select a characteristic fragment ion, which eliminated the interference of non-target ions, making the quantitative inference more accurate and the better repeatability. After the metabolic substance spectrum analysis data of different samples were obtained, the mass spectrometry peaks of all substances were integrated, and the mass spectrometry peaks of the same metabolite in different samples were integrated and corrected [39].

#### 4.3.5. Data Analysis

The original data obtained were preprocessed at first (noise filtering, peak matching and peak extraction) and the data were corrected [40]. Then, the data of quality control entered the stage of statistical analysis.

Statistical analysis used multivariate analysis. Data were log-transformed and mean-centred using SIMCA software (V14.1, MKS Data Analytics Solutions, Umea, Sweden) for PCA and OPLS-DA analysis. PCA analysis was followed by automated modelling analysis [41,42]. The first principal component (PC1) was subjected to OPLS-DA modelling, and the model quality was tested by 7-fold cross validation. After that, the resulting R2Y (the interpretability of the model on the categorical variable Y) and Q2 (the predictability of the model) were used to evaluate the validity of the model. The permutation test was performed multiple times to generate different random Q2 values, which were used to further test model validity [43].

R software (version 3.0.3) was used for Hierarchical clustering analysis (HCA) analysis. The data were log 2 transformed and similarity assessment for clustering was based on the Euclidean distance coefficient.

Based on OPLS-DA analysis, the differential metabolites were screened by the following criteria: (1) If the difference of metabolites content between the control group and the experimental group is more than 2 times or less than 0.5, the difference is considered to be significant; (2) On the basis of the above, the metabolites with VIP ≥ 1 are selected.

Finally, the Kyoto Encyclopaedia of Genes and Genomes (KEGG) Pathway database (http://www.kegg.jp/kegg/pathway.html) [44] and MW-database (Metware Biotechnology Co., Ltd, Wuhan, China) were centred on metabolic reactions and concatenates possible metabolic pathways.

### 4.4. Enzyme Activity Determination

The enzyme activities of 10 enzymes were determined by enzyme-linked immunosorbent assay (ELISA). Leaves were ground to powder in liquid nitrogen and accurately weighed 0.1 g sample in the centrifuge tube. Then, added the same volume of 0.1 mol/L precooled PBS solution, centrifuged about 3000 rpm/min at low-temperature and low-speed for 10 min. 1 mL of the supernatant was collectedn to test. The ELISA detection kit manual of the corresponding enzyme (ProNets Biotechnology Co., Ltd., Wuhan, China) was used to determine the enzyme.

### 4.5. Total RNA Extraction and qRT-PCR Analysis

The total RNA of *C. sinense* ‘Red Sun’ was extracted by RNA extraction kit (TIANGEN, Biotech, Beijing, China). The content of RNA in three samples was determined by nanodrop2000 (Thermo Fisher, Waltham, MA, USA). 500 ng RNA was taken from each sample for reverse transcription by HiScript Q RT SuperMix for qPCR kit (Vazyme Biotech Co., Nanjing, China) to obtain cDNA. cDNA dilution of 10-folds was used for qRT-PCRanalysis.

QRT-PCR used CFX96TM Real-Time System (Bio-Rad, Hercules, CA, USA) following the instructions based on ChamQ SYBR qPCR Master Mix kit (Vazyme Biotech Co., Nanjing, China). CsACTIN was used as normalization standard for gene expression. The gene expression was calculated by 2^−ΔΔCT^. The primers for qRT-PCR are listed in Appendix A.

## 5. Conclusions

In this study, a LC-MS-based metabolomics approach was used to evaluate the difference in metabolites during the change of leaf color of *C. sinense* ‘Red Sun’. This is the first metabolomics study on *C. sinense*. A total of 196 flavonoid-related metabolites were detected. 42 metabolites were identified as differential metabolites during the process of leaf color change. In anthocyanin biosynthetic pathway, 15 metabolites were identified and the contents of them all showed decrease. Especially the contents of Naringenin chalcone, an important intermediate in anthocyanin synthesis, and two important anthocyanins metabolites, Pelargonidin O-acetylhexoside and Anthocyanin 3-O-beta-D-glucoside, decreased to zero in the green leaf stage. The enzyme activity of 10 enzymes related to anthocyanin synthesis showed different change patterns, while the expression of corresponding encoding genes was all down-regulated in the process of leaf color change. Overall, this study substantially contributes to the knowledge flavonoid-related metabolites composition in *C. sinense* and provides important reference values for breeders to improve the leaf color of *C. sinense.*

## Figures and Tables

**Figure 1 ijms-21-01869-f001:**
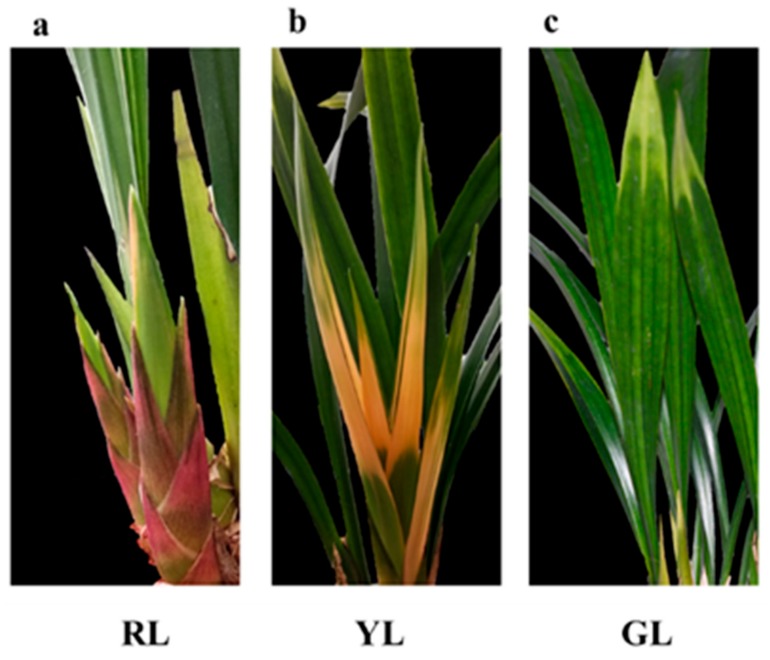
Phenotypes of *Cymbidium sinense* ‘Red Sun’ red leaves (RL) (**a**), yellow leaves (YL) (**b**) and green leaves (GL) (**c**).

**Figure 2 ijms-21-01869-f002:**
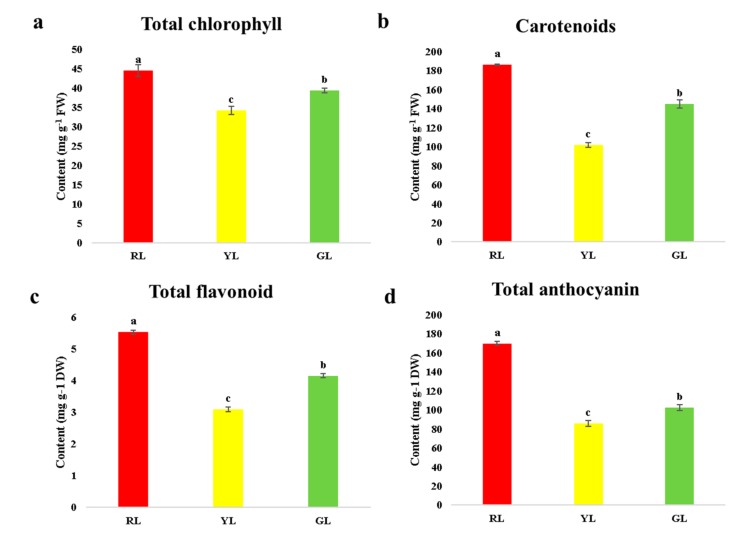
Chlorophyll (**a**), carotenoids content (**b**), total flavonoid (**c**) and total anthocyanin (**d**) content of different color leaves of *Cymbidium sinense* ‘Red Sun’. Bars represent the mean of three biological replicates ±SE. Lowercase letters indicate significant differences at *p* < 0.05.

**Figure 3 ijms-21-01869-f003:**
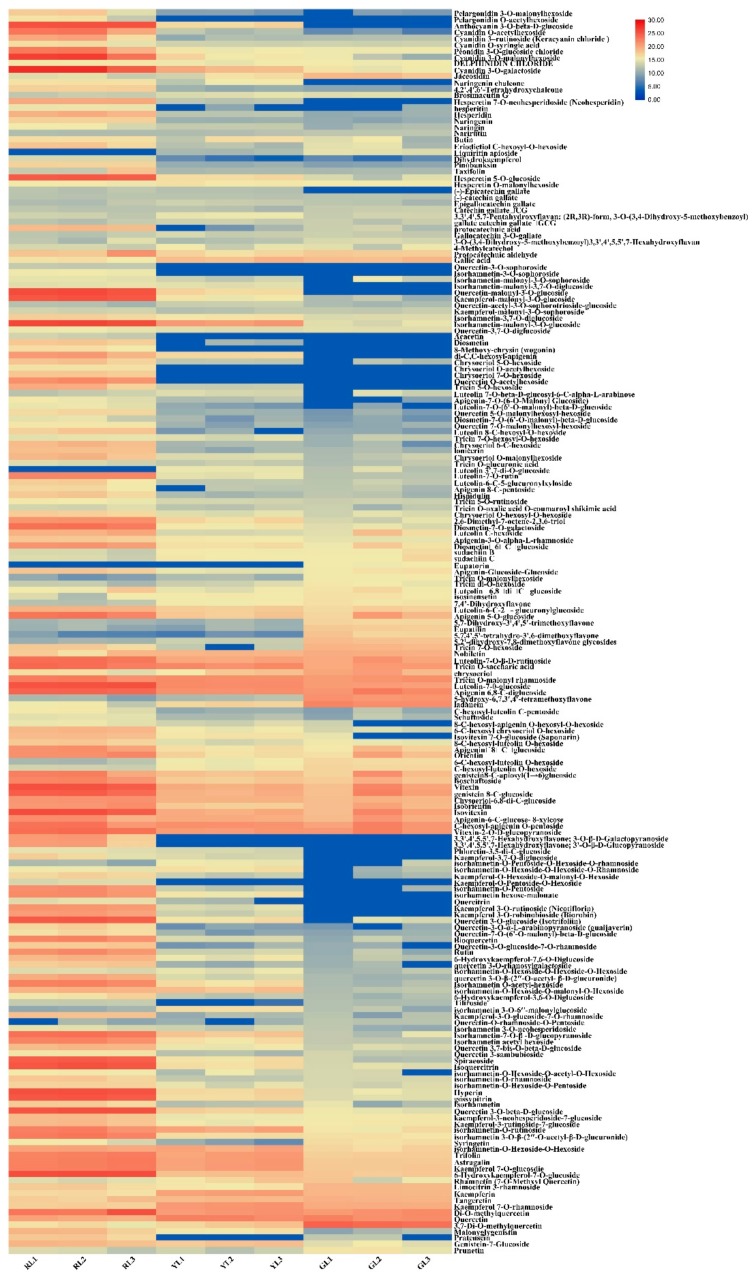
Hierarchical clustering analysis of all metabolites detected in this study. The abscissa indicates three biological replicates of red leaves (RL1, RL2, and RL3), yellow leaves (YL1, YL2, and YL3) and green leaves (GL1, GL2, and GL3), and the ordinate indicates the metabolites detected in this study. The red segments indicate a relatively high content of metabolites, while the blue segments indicate a relatively low content of metabolites. The relative metabolite contents represented by color segments at the corresponding locations are listed in Appendix A.

**Figure 4 ijms-21-01869-f004:**
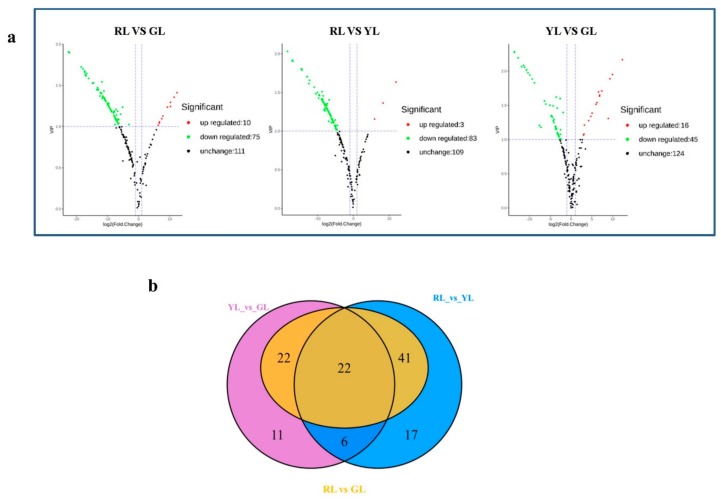
(**a**) Volcano plot of differential metabolites for RL vs. GL, RL vs. YL and YL vs. GL. The colors of the scatter points in Figure 4a indicate the final screening results: red indicates metabolites that were significantly up-regulated; green indicates metabolites that were significantly down-regulated; grey indicates metabolites with no significant difference. (**b**) Venn diagram analysis of differential metabolites. RL, red leaves; YL, yellow leaves; GL, green leaves.

**Figure 5 ijms-21-01869-f005:**
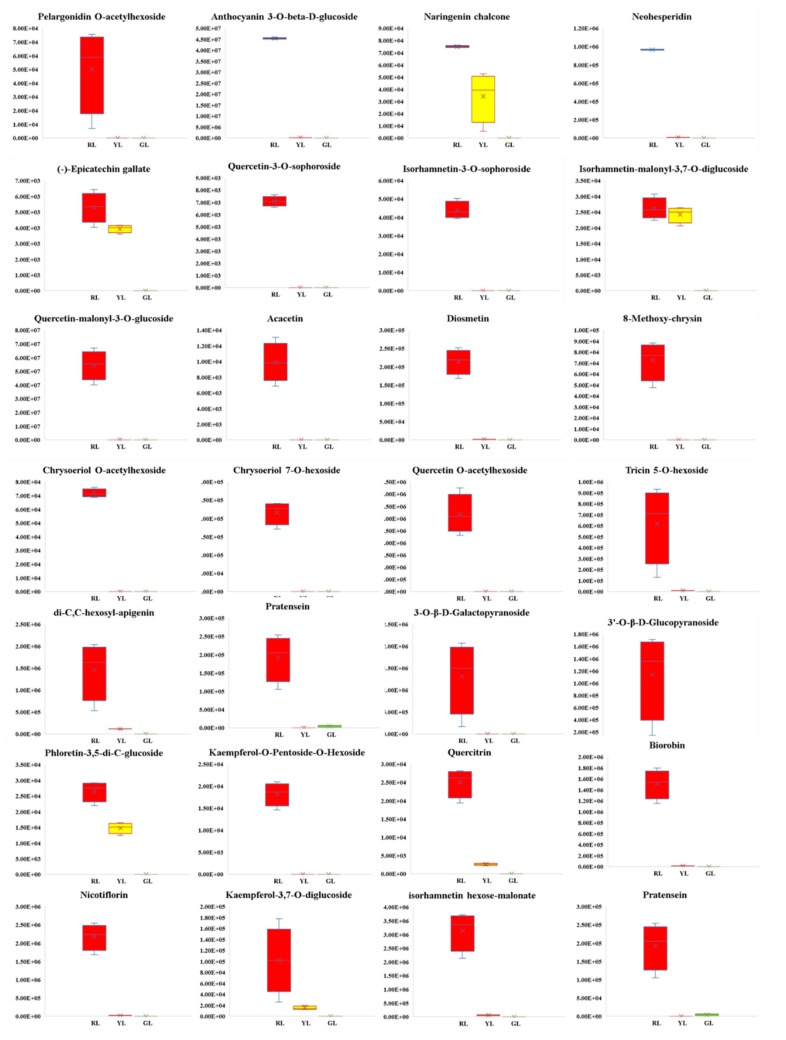
Differences in the content of 28 metabolites in the process of leaf color change. Y-scale represent the integral value of chromatographic peak area.

**Figure 6 ijms-21-01869-f006:**
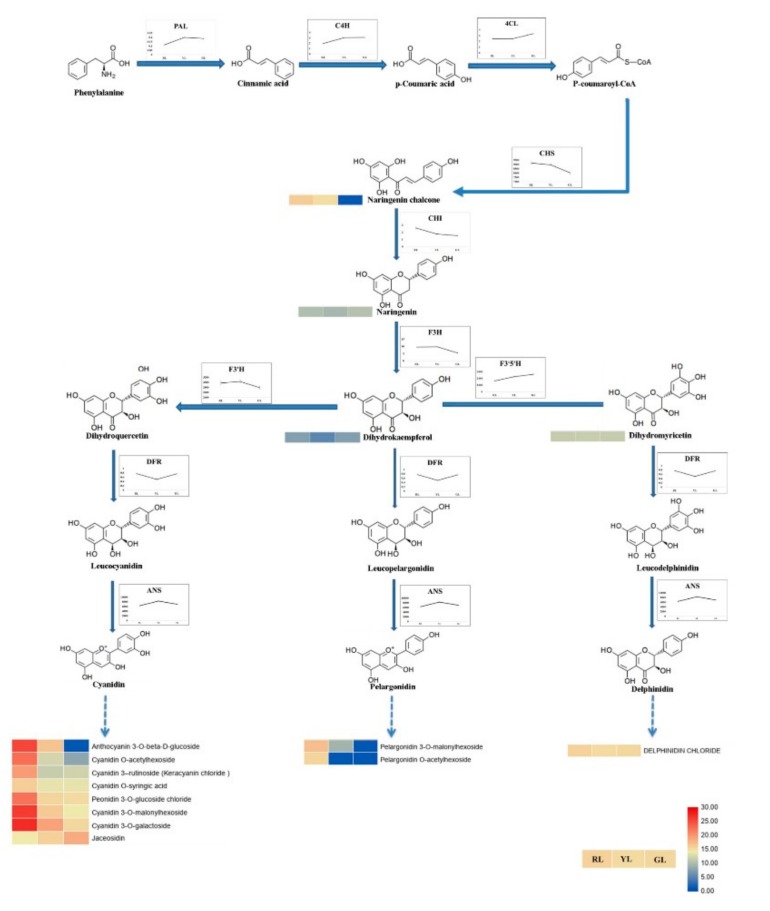
The changing patterns of enzymes activity and intermediate products contents related to anthocyanin synthesis in the process of *Cymbidium sinense* ‘Red Sun’ leaf color change. Red and blue shading in the lower right corner indicates the relatively high-or low content, respectively. PAL, Phenylalanine ammonia lyase, C4H, cinnamate 4-hydroxylase, 4CL, 4-coumarate CoA ligase, CHS, chalcone synthase, CHI, chalcone isomerase, F3H, flavone 3-hydroxylase, F3′H, flavonoid 3′-hydroxylase, F3′5′H, flavonoid 3′,5′-hydroxylase, DFR, dihydroflavonol reductase, ANS, anthocyanidin synthase. Units on y-scale of enzymes activity is U/g.

**Figure 7 ijms-21-01869-f007:**
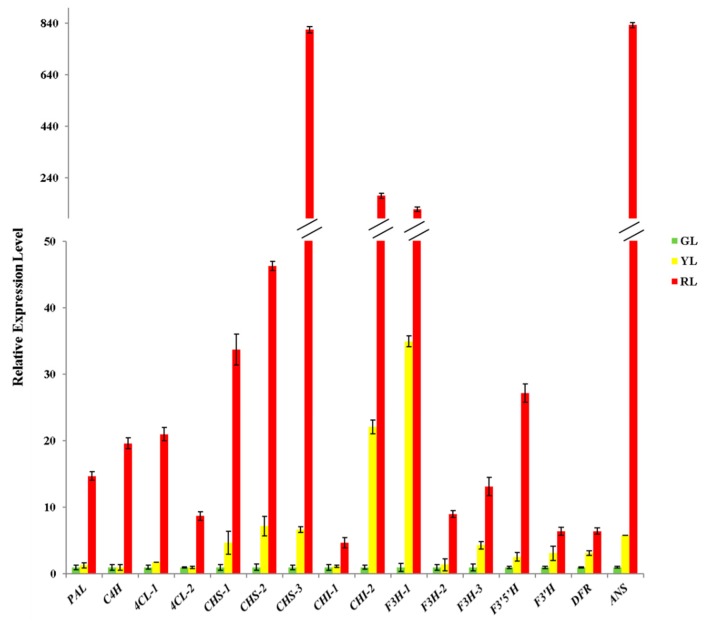
Expression pattern of genes coding enzymes related to anthocyanin synthesis in the process of *Cymbidium sinense* ‘Red Sun’ leaf color change.

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
