# Peer review of "Comparative Metabolomic Analysis Reveals Distinct Flavonoid Biosynthesis Regulation for Leaf Color Development of Cymbidium sinense ‘Red Sun’"

_ijms, 2020, doi:10.3390/ijms21051869_

Round 1
Reviewer 1 Report
The authors present a survey of the development of the Red Sun plant, which is uniquely variagation, specifically with regard to the developmental changes of the plant. This method is carried out using an LC-MS/MS approach, and presents some interesting findings, although the importance of which is not entirely clear in the article. Please see my comments below.
- Abstract: The abstract needs some grammatical editing, but also does not provide enough detail. The authors state that they are looking at "flavonoid metabolites" but do not discuss how they are doing it. Additionally, it is not clear if they mean metabolites of flavonoids, or the class of phytchemicals they are investigating.
- Results: The first section (Section 2.1) states that pigments were not found at the green leaf stage, but this is really strange; shouldn't chlorophyl be present at the least? Figure two, as referenced in the text, shows pigments. I'm not entirely sure what the authors are trying to communicate
- Section 2.2: The authors only address "the UPLC-MS/MS technique..." But what about it? I know this is meant for the materials and methods, but maybe they should at least state the type of UPLC, if they're going to mention it at all in this section.
- Line 101, sentence starting with "The software..." is better placed in the materials and methods.
- Figure 3: The authors show 3 replicates, but do not indicate in the figure legend what they are replicates of. Are they of leaves, different plants... please specify.
- Figure 5 caption: I'm not sure what the authors mean by this figure caption. What is from scratch? for from something to nothing? I think this figure caption needs to be reworded, and an appropriate statistical model should be applied to demonstrate that they are, indeed, different.
- Section 4.1: The authors state that the samples for each stage were taken from different plants, but do not address the replicates as seen in Figure 3.
- Section 4.3.5: How were the data processed? Please provide detail and references.
Author Response
Point 1: The abstract needs some grammatical editing, but also does not provide enough detail. The authors state that they are looking at "flavonoid metabolites" but do not discuss how they are doing it. Additionally, it is not clear if they mean metabolites of flavonoids, or the class of phytchemicals they are investigating.
Response 1: The grammatical has been edited.
The method for looking at “flavonoid-related metabolites” is UPLC-MS/MS- based metabolomics approach. Line 22
The “flavonoid metabolites” in this paper has been all revised to “flavonoid-related metabolites”.
Point 2: The first section (Section 2.1) states that pigments were not found at the green leaf stage, but this is really strange; shouldn't chlorophyll be present at the least? Figure two, as referenced in the text, shows pigments. I'm not entirely sure what the authors are trying to communicate.
Response 2: The phrase has been changed to ‘The contents of all four pigments in green leaves were higher than those in yellow leaves, but lower than those in red leaves (Figure 2).’ Lines 92-94
Point 3: Section 2.2: The authors only address "the UPLC-MS/MS technique..." But what about it? I know this is meant for the materials and methods, but maybe they should at least state the type of UPLC, if they're going to mention it at all in this section.
Line 101, sentence starting with "The software..." is better placed in the materials and methods.
Response 3: The type of UPLC is Shim-pack UFLC SHIMADZU CBM30A, the type of MS/MS is Applied Biosystems 4500 QTRAP. Lines 105-106
The sentence starting with "The software..." have been placed in the materials and methods. Line 363
Point 4: Figure 3: The authors show 3 replicates, but do not indicate in the figure legend what they are replicates of. Are they of leaves, different plants... please specify.
Response 4: The abscissa indicates three biological replicates of red leaves (RL1, RL2 and RL3), yellow leaves (YL1, YL2 and YL3) and green leaves (GL1, GL2 and GL3). Lines 127-130
Point 5: Figure 5 caption: I'm not sure what the authors mean by this figure caption. What is from scratch? for from something to nothing? I think this figure caption needs to be reworded, and an appropriate statistical model should be applied to demonstrate that they are, indeed, different.
Response 5: After a comprehensive analysis of the contents of all the differential metabolites, we found that the contents of 28 metabolites turned to zero in the process of leaf color change, as shown in Figure 5.
The figure caption reworded to “Differences in the content of 28 metabolites in the process of leaf color change”.
The reason why I choose boxplot as the statistical model is because boxplot can measured the dispersion of three biological replicates to provide a better understand of the data for readers.
Point 6: Section 4.1: The authors state that the samples for each stage were taken from different plants, but do not address the replicates as seen in Figure 3.
Response 6: The replicates of samples have been added in section 4.1. Lines 308-310
Point 7: Section 4.3.5: How were the data processed? Please provide detail and references.
Response 7: The details and references have been added in 4.3.5. Lines 381-401
Reviewer 2 Report
The manuscript presented by Gao and co-authors “Comparative metabolomic analysis reveals distinct flavonoid biosynthesis regulation for leaf color development of Cymbidium sinense ‘Red Sun’” investigates the metabolic, enzymatic and gene expression changes in leaves of Cymbidium sinense “Red Sun”. This research is well-structured and the results are of scientific interests. However, the manuscript needs more work and clarification in some points, as I explain into more detail below. I hope help authors to improve the manuscript.
Major:
- Abstract and Conclusions should be deeply revised. I think that these two parts of the manuscript need a little bit of work, avoiding to report a list of number of the metabolites found during the three different phases of leaf development. The authors should clearly report the main metabolic changes observed passing from red leaves to yellow leaves and green leaves. Otherwise the main findings of the work cannot be deduced.
- Figure 5 is not reported in the result section. The authors have to add the description of this important figure. I think a reference to this figure should be added in line 190 and at the end of line 195. However, some more results on the main metabolites should be added in this section.
- I am not really convinced of the colorimetric method utilized to estimate the total amount of chlorophyll, carotenoids, flavonoids and anthocyanins. I don’t think it is possible to report exactly (as mg g-1DW) the amounts of this compounds using a colorimetric kit. In the case it is, the authors should describe in details the standard they have used, the instrument they have used and so on.. In addition I have other two questions on this point: 1) how were the samples dried? 2) why did the authors not used the HPLC data (since they have done a qualitative and quantitative analysis) to report the total amount of flavonoids and anthocyanins? In addition, the total amount of chlorophyll and carotenoids can be measured with very simple, but trustable, spectrophotometric analyses (see for ex. Lichtenthaler, H. K., & Buschmann, C. (2001). Extraction of phtosynthetic tissues: chlorophylls and carotenoids. Current protocols in food analytical chemistry, 1(1), F4-2; Biehler, E., Mayer, F., Hoffmann, L., Krause, E., & Bohn, T. (2010). Comparison of 3 spectrophotometric methods for carotenoid determination in frequently consumed fruits and vegetables. Journal of food science, 75(1), C55-C61).
Minor
- Line 29 (abstract): I think the word “provided” should be changed in “providing”.
- Line 46 (introduction): “the reason”. The authors should be more precise; for ex. they could say “metabolic changes associated with…”
- Line 51 (introduction): the ref [2] is not related to orchid and it is too specific for the context. Similarly, the ref [5] is very specific. They authors should provide evidences of anthocyanins in leaves, eventually also mention the potential role of these compounds (see for ex. Lee, D. W., & Gould, K. S. (2002). Why leaves turn red: pigments called anthocyanins probably protect leaves from light damage by direct shielding and by scavenging free radicals. American Scientist, 90(6), 524-531. Gould, K. S. (2004). Nature's Swiss army knife: the diverse protective roles of anthocyanins in leaves. BioMed Research International, 2004(5), 314-320). In addition, this reference on anthocyanin in Cymbidium could be of interest for the work: Albert, N. W., Arathoon, S., Collette, V. E., Schwinn, K. E., Jameson, P. E., Lewis, D. H., ... & Davies, K. M. (2010). Activation of anthocyanin synthesis in Cymbidium orchids: variability between known regulators. Plant Cell, Tissue and Organ Culture (PCTOC), 100(3), 355-360.
- Line 89 (results): “no significant amounts were observed at the green leaf stage”. This sentence is without a clear meaning. The amount of the different compounds are reported also in green leaves. I suggest to explain or rephrase this sentence.
- Units on y-scale of Fig. 5 and 7 should be added.
- The plots reported in Fig. 6 are very small, so I suggest to add the units utilized for y-scale in the text of the caption.
- the molecule anthocyanin 3-O-beta –D-glucoside, which has not been completely identified should be reported in the results mentioning its molecular weight.
- Line 221 (discussion): “the mechanism of leaf color regulation from the perspective of small molecular..”. This sentence should be rephrased because there is a large body of evidences, using metabolomics, to study this mechanism. These are just some citations I have found on this point:
Nakabayashi, R., Kusano, M., Kobayashi, M., Tohge, T., Yonekura-Sakakibara, K., Kogure, N., ... & Takayama, H. (2009). Metabolomics-oriented isolation and structure elucidation of 37 compounds including two anthocyanins from Arabidopsis thaliana. Phytochemistry, 70(8), 1017-1029.
Yamazaki, M., Nakajima, J. I., Yamanashi, M., Sugiyama, M., Makita, Y., Springob, K., ... & Saito, K. (2003). Metabolomics and differential gene expression in anthocyanin chemo-varietal forms of Perilla frutescens. Phytochemistry, 62(6), 987-995.
Kim, Y. B., Park, S. Y., Thwe, A. A., Seo, J. M., Suzuki, T., Kim, S. J., ... & Park, S. U. (2013). Metabolomic analysis and differential expression of anthocyanin biosynthetic genes in white-and red-flowered buckwheat cultivars (Fagopyrum esculentum). Journal of agricultural and food chemistry, 61(44), 10525-10533.
Rothenberg, D. O. N., Yang, H., Chen, M., Zhang, W., & Zhang, L. (2019). Metabolome and transcriptome sequencing analysis reveals anthocyanin metabolism in pink flowers of anthocyanin-rich Tea (Camellia sinensis). Molecules, 24(6), 1064.
Yang, B., He, S., Liu, Y., Liu, B., Ju, Y., Kang, D., ... & Fang, Y. (2020). Transcriptomics Integrated with Metabolomics Reveals the Effect of Regulated Deficit Irrigation on Anthocyanin Biosynthesis in Cabernet Sauvignon Grape Berries. Food Chemistry, 126170.
Zhou, S., Chen, J., Lai, Y., Yin, G., Chen, P., Pennerman, K. K., ... & Wang, C. (2019). Integrative analysis of metabolome and transcriptome reveals anthocyanins biosynthesis regulation in grass species Pennisetum purpureum. Industrial Crops and Products, 138, 111470.
Author Response
Point 1: Abstract and Conclusions should be deeply revised. I think that these two parts of the manuscript need a little bit of work, avoiding to report a list of number of the metabolites found during the three different phases of leaf development. The authors should clearly report the main metabolic changes observed passing from red leaves to yellow leaves and green leaves. Otherwise the main findings of the work cannot be deduced.
Response 1: Abstract and Conclusions have been deeply revised. The main metabolic changes observed passing from red leaves to yellow leaves and green leaves have been highlighted in lines 25-27 and lines 425-427 .
Point 2: Figure 5 is not reported in the result section. The authors have to add the description of this important figure. I think a reference to this figure should be added in line 190 and at the end of line 195. However, some more results on the main metabolites should be added in this section.
Response 2: Figure 5 have been reported in lines 172-177.
The reference has been added in line 196.
More details about the main metabolites have been added in lines 198-204.
Point 3: I am not really convinced of the colorimetric method utilized to estimate the total amount of chlorophyll, carotenoids, flavonoids and anthocyanins. I don’t think it is possible to report exactly (as mg g-1DW) the amounts of this compounds using a colorimetric kit. In the case it is, the authors should describe in details the standard they have used, the instrument they have used and so on. In addition I have other two questions on this point: 1) how were the samples dried? 2) why did the authors not used the HPLC data (since they have done a qualitative and quantitative analysis) to report the total amount of flavonoids and anthocyanins? In addition, the total amount of chlorophyll and carotenoids can be measured with very simple, but trustable, spectrophotometric analyses (see for ex. Lichtenthaler, H. K., & Buschmann, C. (2001). Extraction of phtosynthetic tissues: chlorophylls and carotenoids. Current protocols in food analytical chemistry, 1(1), F4-2; Biehler, E., Mayer, F., Hoffmann, L., Krause, E., & Bohn, T. (2010). Comparison of 3 spectrophotometric methods for carotenoid determination in frequently consumed fruits and vegetables. Journal of food science, 75(1), C55-C61).
Response 3: I rechecked the method for four pigment contents analysis. The contents of total chlorophyll and carotenoids were determined by spectrophotometric analyses, and the contents of total flavonoids and anthocyanin were indeed determined by colorimetry.
The samples were dried to a constant weight in a drying oven. Line 319
The data of four pigments contents were observed before the metabolomics analysis, it help us to assess the difference of total flavonoids and anthocyanins during leaf color change.
Point 4: Line 29 (abstract): I think the word “provided” should be changed in “providing”.
Response 4: The word “provided” has been changed in “providing”. Line 31
Point 5: Line 46 (introduction): “the reason”. The authors should be more precise; for ex. they could say “metabolic changes associated with…”
Response 5: This sentence has been changed to ‘So far, the metabolic changes associated with the formation of red leaves in C. sinense ‘Red Sun’ is not known.’ Lines 48-49
Point 6: Line 51 (introduction): the ref [2] is not related to orchid and it is too specific for the context. Similarly, the ref [5] is very specific. They authors should provide evidences of anthocyanins in leaves, eventually also mention the potential role of these compounds (see for ex. Lee, D. W., & Gould, K. S. (2002). Why leaves turn red: pigments called anthocyanins probably protect leaves from light damage by direct shielding and by scavenging free radicals. American Scientist, 90(6), 524-531. Gould, K. S. (2004). Nature's Swiss army knife: the diverse protective roles of anthocyanins in leaves. BioMed Research International, 2004(5), 314-320). In addition, this reference on anthocyanin in Cymbidium could be of interest for the work: Albert, N. W., Arathoon, S., Collette, V. E., Schwinn, K. E., Jameson, P. E., Lewis, D. H., ... & Davies, K. M. (2010). Activation of anthocyanin synthesis in Cymbidium orchids: variability between known regulators. Plant Cell, Tissue and Organ Culture (PCTOC), 100(3), 355-360.
Response 6: The evidences of anthocyanins in leaves, the potential role of these compounds and reference on anthocyanin in Cymbidium have been mentioned in Lines 53-56.
Point 7: Line 89 (results): “no significant amounts were observed at the green leaf stage”. This sentence is without a clear meaning. The amount of the different compounds are reported also in green leaves. I suggest to explain or rephrase this sentence.
Response 7: The explanation has been rephrased in Lines 93-94.
Point 8: Units on y-scale of Fig. 5 and 7 should be added.
Response 8: Unis on y-scale of Fig. 5 has been added in the text of the caption. Unis on y-scale of Fig. 7 has been added in the figure.
Point 9: The plots reported in Fig. 6 are very small, so I suggest to add the units utilized for y-scale in the text of the caption.
Response 9: Units on y-scale of Fig. 6 has been added in the text of the caption.
Point 10: The molecule anthocyanin 3-O-beta –D-glucoside, which has not been completely identified should be reported in the results mentioning its molecular weight.
Response 10: The molecular weight of 3-O-beta –D-glucoside is 449.089 Da. Line 203
Point 11: Line 221 (discussion): “the mechanism of leaf color regulation from the perspective of small molecular.”. This sentence should be rephrased because there is a large body of evidences, using metabolomics, to study this mechanism. These are just some citations I have found on this point:
Nakabayashi, R., Kusano, M., Kobayashi, M., Tohge, T., Yonekura-Sakakibara, K., Kogure, N., ... & Takayama, H. (2009). Metabolomics-oriented isolation and structure elucidation of 37 compounds including two anthocyanins from Arabidopsis thaliana. Phytochemistry, 70(8), 1017-1029.
Yamazaki, M., Nakajima, J. I., Yamanashi, M., Sugiyama, M., Makita, Y., Springob, K., ... & Saito, K. (2003). Metabolomics and differential gene expression in anthocyanin chemo-varietal forms of Perilla frutescens. Phytochemistry, 62(6), 987-995.
Kim, Y. B., Park, S. Y., Thwe, A. A., Seo, J. M., Suzuki, T., Kim, S. J., ... & Park, S. U. (2013). Metabolomic analysis and differential expression of anthocyanin biosynthetic genes in white-and red-flowered buckwheat cultivars (Fagopyrum esculentum). Journal of agricultural and food chemistry, 61(44), 10525-10533.
Rothenberg, D. O. N., Yang, H., Chen, M., Zhang, W., & Zhang, L. (2019). Metabolome and transcriptome sequencing analysis reveals anthocyanin metabolism in pink flowers of anthocyanin-rich Tea (Camellia sinensis). Molecules, 24(6), 1064.
Yang, B., He, S., Liu, Y., Liu, B., Ju, Y., Kang, D., ... & Fang, Y. (2020). Transcriptomics Integrated with Metabolomics Reveals the Effect of Regulated Deficit Irrigation on Anthocyanin Biosynthesis in Cabernet Sauvignon Grape Berries. Food Chemistry, 126170.
Zhou, S., Chen, J., Lai, Y., Yin, G., Chen, P., Pennerman, K. K., ... & Wang, C. (2019). Integrative analysis of metabolome and transcriptome reveals anthocyanins biosynthesis regulation in grass species Pennisetum purpureum. Industrial Crops and Products, 138, 111470.
Response 11: This sentence has been rephrased to ‘the mechanism of leaf color regulation from the perspective of small molecular metabolites needs to be studied further’. Lines 229-230
Round 2
Reviewer 1 Report
- Page 1, line 24: Sentence that begins with “42...” typically, numbers should not start a sentence
- Page 1, lines 31- 33, Sentence beginning with “The change pattern of enzyme activity…” I think the authors are trying to indicate that the enzymes for anthocyanin synthesis was down regulated in green leaf, agreeing with the previous statement, but it is difficult to understand what the intent is behind the sentence.
- Page 2, line 84, sentence beginning with “Combine…” is not properly worded, please check the grammar and rephrase
Author Response
Point 1: Line 22: the word “metabolite” should be deleted.
Response 1: The word “metabolite” has been deleted. Line 22
Point 2: Line 24: Please rephrase the sentence beginning with “ A total of 42 metabolites….”
Response 2: The sentence has been changed to ‘In anthocyanin biosynthetic pathway, content of all 15 identified metabolites showed downregulation trend in the process of leaf color change’. Lines 24-25
Point 3: Line 27: Please specify in which changing process: from red to yellow or from yellow to green.
Response 3: The changing process has been specified: from red leaves turn to yellow and finally to green. Line 30
Point 4: Figure 2b: * “carotenoids”
Response 4: The word ‘carotenoid’ has been changed to ‘carotenoids’. Line 103
Point 5: Line 252-257: This part is interesting but the three sentences are very repetitive. I suggest to rephrase this part in order to give this information in a more concise and vivid way.
Response 5: These three sentences have been rephrased to ‘Based on wide target metabolomics analysis,only 6 and 15 differential flavonoid-related metabolites were detected in tea leaves and ginkgo biloba leaves, respectively . Based on phenolic-targeted secondary metabolites analysis in purple fig peel, only 15 differential flavonoid-related metabolites (including four anthocyanins metabolites) were detected ’. Lines 244-252.
Point 6: Line 266: Please change “according” with “In accordance with the results…”
Response 6: The sentence has been changed to ‘In accordance with the results…’. Line 259
Point 7: Line 275: Please delete “In terms of types”.
Response 7: “In terms of types” has been deleted. Line 268
Point 8: Line 279-281: This sentence is not clear: “the content of anthocyanins showed high content in purple petals but zero in white petals of Phalaenopsis are the derivatives of six kinds of cyanidin and a kind of delphinidin”. Please rephrase.
Response 8: This sentence has been changed to ‘The content of derivatives of six kinds of cyanidin and a kind of delphinidin was high in purple petals but zero in white petals of Phalaenopsis’. Lines 272-273
Point 9: Line 331-333: There is still something wrong in this part. Chlorophyll and carotenoids are not soluble in water. If you have extracted these metabolites in this way the method is wrong and untrustable. Please check carefully the utilized method (solvent of extraction, etc) and specify the wavelengths and the formula used to calculate these metabolites.
Response 9: See Lines 323-334.
Point 10: Line 335: Please specify the temperature of the oven.
Response 10: The temperature is 37℃.Line 356
Point 11: Line 336-337: Please add more details on the extraction of these metabolites.
Response 11: See Lines 338-371.
Point 12: Line 450: “In anthocyanin biosynthesis pathway” should be changed in “In anthocyanin biosynthetic pathway”.
Response 12:“In anthocyanin biosynthesis pathway” has been changed in “In anthocyanin biosynthetic pathway”. Line 473
Point 13: Line 453: delete “all”.
Response 13: ‘all’ has been deleted. Line 476
Reviewer 2 Report
The manuscript has been improved. However, I suggest some further corrections before its publication.
Line 22: the word “metabolite” should be deleted.
Line 24: Please rephrase the sentence beginning with “ A total of 42 metabolites….”
Line 27: Please specify in which changing process: from red to yellow or from yellow to green.
Figure 2b: * “carotenoids”
Line 252-257: This part is interesting but the three sentences are very repetitive. I suggest to rephrase this part in order to give this information in a more concise and vivid way.
Line 266: Please change “according” with “In accordance with the results…”
Line 275: Please delete “In terms of types”.
Line 279-281: This sentence is not clear: “the content of anthocyanins showed high content in purple petals but zero in white petals of Phalaenopsis are the derivatives of six kinds of cyanidin and a kind
of delphinidin”. Please rephrase.
Line 331-333: There is still something wrong in this part. Chlorophyll and carotenoids are not soluble in water. If you have extracted these metabolites in this way the method is wrong and untrustable. Please check carefully the utilized method (solvent of extraction, etc) and specify the wavelengths and the formula used to calculate these metabolites.
Line 335: Please specify the temperature of the oven.
Line 336-337: Please add more details on the extraction of these metabolites.
Line 450: “In anthocyanin biosynthesis pathway” should be changed in “In anthocyanin biosynthetic pathway”.
Line 453: delete “all”.
Author Response
Point 1: Page 1, line 24: Sentence that begins with “42...” typically, numbers should not start a sentence
Response 1: This sentence has been changed to ‘In the process of leaf color change, 42 metabolites were identified as having significantly different contents and the content of 28 differential metabolites turned to zero’. Lines 22-24
Point 2: Page 1, lines 31- 33, Sentence beginning with “The change pattern of enzyme activity…” I think the authors are trying to indicate that the enzymes for anthocyanin synthesis was down regulated in green leaf, agreeing with the previous statement, but it is difficult to understand what the intent is behind the sentence.
Response 2: This sentence has been changed to ‘The change pattern of enzyme activity of 10 enzymes involved in anthocyanin biosynthetic pathway showed different trend from red leaves turn to yellow and finally to green, while the expression of genes encoding these enzymes was all down-regulated in the process of leaf color change’. Lines 29-32
Point 3: Page 2, line 84, sentence beginning with “Combine…” is not properly worded, please check the grammar and rephrase
Response 3: This sentence has been changed to ‘Physiology, enzyme kinetics and molecular biology experiments were carried out to explore the mechanism of leaf color difference’. Lines 85-86